# Resilience Evaluation of High-Speed Railway Subgrade Construction Systems in Goaf Sites

Hui Wang [1,2], Jing Zhou [1], Zhiyuan Dun [3,*], Jianhua Cheng [1,2], Hujun Li [1] and Zhilin Dun [1,2]

[1] School of Civil Engineering, Henan Polytechnic University, Jiaozuo 454003, China; wanghui9962@hpu.edu.cn (H.W.); zhoujing210508@163.com (J.Z.); cheng15@hpu.edu.cn (J.C.); lihujunsr@outlook.com (H.L.); dzl1964@163.com (Z.D.)
[2] Henan Engineering Research Center for Ecological Restoration and Construction Technology of Goaf Sites, Jiaozuo 454003, China
[3] School of Civil and Architectural Engineering, Jiaozuo University, Jiaozuo 454000, China
* Correspondence: dzy207@163.com; Tel.: +86-18739160937

**Abstract:** When the high-speed railway construction project passes through goaf sites, the uncertain impacts from internal and external environments faced by the system are gradually characterized by complexity and variability, and the disastrous consequences are becoming increasingly prominent. The risk resistance ability and accident recovery ability of the high-speed railway subgrade construction system are crucial for improving the safety management level of the construction site. Based on the concept of resilience, this paper discusses the connotation of resilience applicable to the construction system of high-speed railway foundations in goaf sites, and an evaluation index system including 25 indicators is constructed. Then, the resilience evaluation model is constructed by ANP, entropy weight method and fuzzy comprehensive evaluation method. Taking the construction system of Taijiao high-speed railway subgrade in the underlying goaf as an example, this model is verified. The verification results show that the grade of construction system is II (high resilience). The evaluation result is consistent with the actual engineering situation, and the evaluation model is effective. It can be used as a theoretical basis for safety management of high-speed railway construction projects, and a full process analysis method based on resilience theory is established.

**Keywords:** goaf site; high-speed railway; sustainability; resilience evaluation; ANP-entropy weight-fuzzy evaluation model

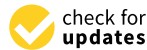

## 1. Introduction

At present, with the rapid development of the Chinese high-speed railway network, some critical routes have to cross the mined-out area, resulting in frequent safety accidents in the construction of the high-speed railway foundation above the goaf. This is because, in the process of the high-speed railway crossing over the goaf, the rock mass is destroyed, the caved belt, fault belt and caving belt are deformed, the roof of the goaf is unstable, and the rock mass destruction is transmitted upward. Finally, the instability of the goaf leads to the deformation of the roadbed above it, resulting in disasters. This type of disaster has the characteristics of coupling chains (as shown in Figure 1), which also poses a serious challenge to the safe construction and operation of the high-speed railway. For instance, in China, the Handan-Changzhi high-speed railway, Jilin-Huichun high-speed railway and the No. 70 high-speed railway in the Midwest of the United States all pass through the goaf sites, the foundation activation in the goaf leads to the settlement and uneven deformation of the railway roadbed, which poses a great threat to the safety of the high-speed railway [1–3]. Therefore, safety is the primary issue of high-speed railway construction in the mined-out area. The existing research on the construction safety of high-speed railway foundations is often focused on the prevention and control of accidents on the construction site, and the research on the post-recovery and optimization capability of

the system is insufficient [4]. There are many risk factors in high-speed railway construction engineering, resilience is well chosen to represent the risk defense ability and recovery ability of the system after disturbance [5]. Considering the results and identified limitations and gaps of previous studies, it is of great significance to introduce the resilience theory into the safety management of high-speed railway foundation construction systems in goaf sites to improve the resilience and risk resistance of the safety system on the construction site. It is also an important way for human society and economies to achieve sustainability and build a resource-saving and environment-friendly society.

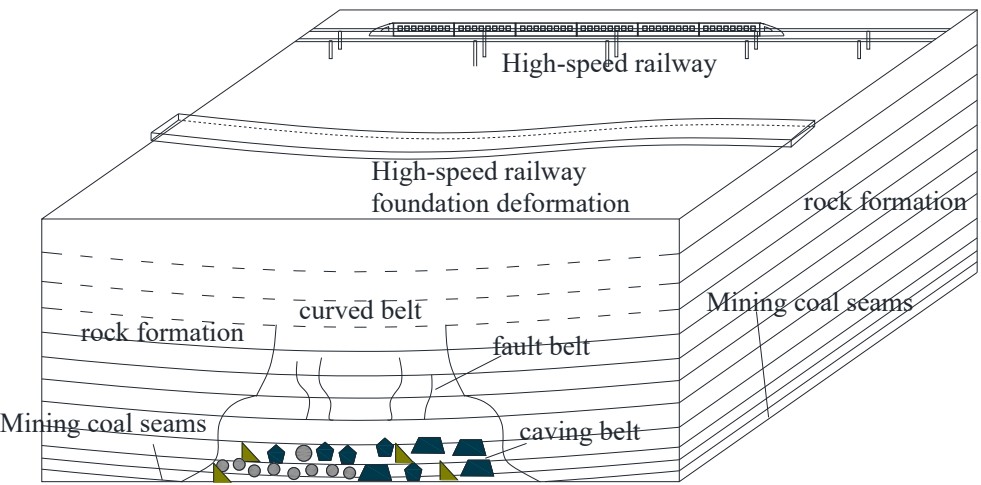

**Figure 1.** Schematic diagram of coupled chain disaster in goaf.

Since the 21st century, under the background of the greenhouse effect and global climate change, the theory of sustainable development has become one of the research focuses. The concept of resilience not only considers the influence of internal and external conditions of the system, but also brings human activities into the resilience evaluation. It not only reflects the understanding and comprehension of humans to resources and the environment system, but also shows the relationship between human activities and sustainable development. Existing research [6–10] has defined resilience in terms of engineering resilience, system safety resilience and management resilience, but the essence is the same, they all emphasize the ability of the system to withstand, absorb, recover and improve from external shocks and disturbances, which provides a theoretical basis for the research content of this paper. There is conceptual confusion and overlap between the elements of resilience, and there is no clear boundary. The constituent elements of resilience are expressed as a set of concepts, including risk, adaptability, resilience and elasticity. According to these studies, we concluded the resilience of high-speed railway subgrade construction systems in goaf sites refers to that in the construction process of high-speed railway subgrade above goaf, and that, in the face of uncertain risks or disturbances, the system can maintain its stability and adapt to risks as soon as possible, and make up for the situation that the system cannot be used normally due to risk impact through the safety backup function. By learning from experience, the system can respond, learn and recover from the risks quickly when the next risk accident occurs. The resilience management of the high-speed railway system is conducive to the efficient management and sustainable development of the system and meets the requirements of safety management theory.

The first step of the resilience evaluation is to build a resilience index system. In China, Hao [10], Huang [11], Zhao [9], Bai [12] and Li Tongyue [13] constructed a resilience index system with four typical resilience characteristics of stability, redundancy, efficiency and adaptability; Jiang [14] divided the vulnerability of the emergency management system into three parts by introducing vulnerability theory—exposure, adaptability and recovery—and constructed offensive and defensive index systems according to the characteristics of emergency management systems in the hydropower engineering construction stage.

Chen [15] identified the influencing factors of system vulnerability in the construction stage of green building projects from three aspects: system exposure, system adaptability and system sensitivity; Zhong [16] constructed a resilience evaluation index system based on three characteristic capabilities of highway tunnel engineering construction system resilience and construction subsystems. In foreign research, Leire [17], Bruneau [18], Zobel [19], Gibson and Tarrant [20] broke resilience down into four dimensions: technical resilience, organizational resilience, economic resilience and social resilience. Ahern [21] discussed the theory of resilience as it applies to urban conditions, and offered a suite of strategies intended to build urban resilience capacity: multifunctionality, redundancy and modularization, (bio and social) diversity, multi-scale networks and connectivity, and adaptive planning and design.

In terms of resilience evaluation methods, scholars generally adopt the network analytic hierarchy process (ANP) [22–25], TOPSIS method [26–31], entropy weight method [9,32–35], cloud model [36–40] and Bayesian network [41–44] to construct a resilience evaluation model.

Combined with the theory of safety science [11], socio-technical systems [45], and the four elements theory of accidents [10,46], considering that the construction system is affected by personnel, machinery, environment, management, and other factors [47,48], the high-speed railway subgrade construction system in goaf sites is divided into four subsystems: organization member system, material technology system, management system, and environmental system.

It can be seen from the above literature that these scholars' research on the construction of a resilience evaluation index system and evaluation method involves many aspects, including the resilience evaluation of the urban resilience level, subway tunnel resilience level, and the resilience level of floods and high-speed railway operation disasters. However, these results cannot be directly applied to the resilience evaluation of all systems, especially the resilience evaluation of high-speed railway subgrade construction systems in mined-out areas.

In summary, according to the safety management theory and resilience theory, we plan to put forward the connotation and analysis of the characteristic elements of resilience of high-speed railway subgrade construction systems based on the characteristics of the goaf site. The resilience indexes of the system will also be emphatically analyzed. Besides, we will explore the influence level of the resilience indexes on the system by the ANP-entropy weight-fuzzy comprehensive evaluation model, and expound on the basic situation that affects the resilience of the construction system of Taijiao Railway to determine its resilience level, to provide new ideas and methods for the study of similar construction system safety management. The comprehensive research framework is drawn according to the overall research idea, as shown in Figure 2.

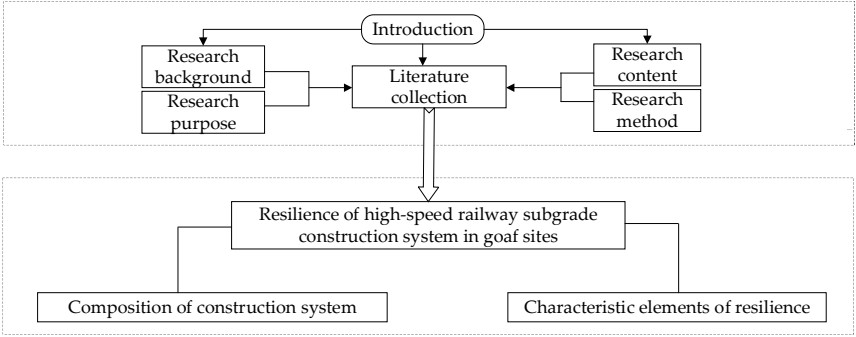

**Figure 2.** *Cont.*

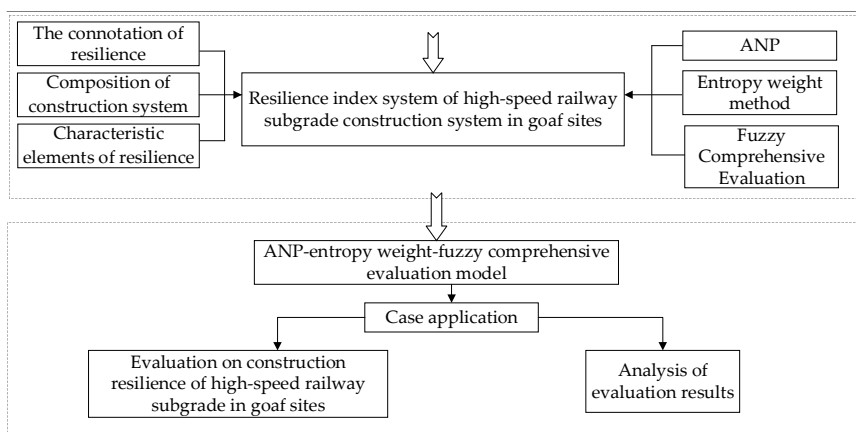

**Figure 2.** Comprehensive research framework.

## 2. Resilience of High-Speed Railway Subgrade Construction Systems in Goaf Sites

*2.1. Composition of High-Speed Railway Subgrade Construction Systems in Goaf Sites*

The construction of high-speed railway subgrade in goaf sites is a complex system, which is composed of personnel, materials, administration, and the environment in the construction process. According to the previous research results [9,10,16], it was divided into an organization member system, material technology system, management system, and environmental system.

The organization member system includes personnel's safety cognitive ability, business ability, and physical and mental state. In the face of the risks and challenges brought by the severe environment, operators in the high-speed rail construction project above the goaf should have sufficient safety awareness to reduce or avoid safety problems during construction; furthermore, the lack of professional and technical personnel also bring risks to the construction process.

The material technology system includes mechanical equipment, material, technical scheme, emergency facilities, and the subgrade deformation monitoring system. This is because in the goaf site, many factors such as the thickness of overburden, the nature of rock and soil, the degree of rock fragmentation, hydrological conditions, the depth of goaf, the collapse, the deformation of subgrade, and the size of engineering quantity affect the selection of mechanical equipment. The goaf treatment scheme is one of the key technologies in the construction of mining area engineering, which affects the quality, duration, and cost of construction [49]. In addition, materials, technologies, and safeguards will be considered to ensure production safety [10].

The management system mainly includes rules, regulations, and emergency management. The complex and changeable environment of the goaf site and the particularity of high-speed railway construction make accidents unpredictable and sudden. Therefore, it is necessary to prepare for emergency prevention, establish a safety inspection team, and effectively control the possible safety problem before the accident occurs, which can fundamentally make the construction personnel have good safety consciousness, and implement the safety rules and regulations [50]. Afterward, it can formulate emergency treatment measures timely to ensure that the risk can be handled and eliminated with the fastest speed and the least investment after it occurs [51], and report the risk per the procedure to avoid causing greater impact.

In the environmental system, factors such as the characteristics of goaf, site operation environment, environmental risk assessment and response, and other factors are studied. To ensure the safety of the high-speed railway construction in goaf, considering the influence of the activation deformation of the mined-out area foundation and the uneven settlement of the subgrade on the construction of the high-speed railway, Mu Wenguang [52], Li Guohe [53] and other scholars [54] analyzed the influence range of goaf on the high-speed railway using INSAR interpretation and geophysical exploration, and studied the stability

of railway goaf sites. Therefore, the analysis of the characteristics and working environment of the goaf in the construction site is conducted to evaluate the stability of the site, assess and respond to risks, and ensure construction safety.

### 2.2. Characteristic Elements of Resilience

In the previous descriptions of the concept of resilience, the ability to resist and absorb risks to maintain its stability is emphasized, and ultimately to adapt to risks and restore the normal operation of the system. Therefore, this paper describes the characteristics of resilience from two aspects of risk resistance and recovery capability based on the concept of sustainability.

### 2.2.1. Risk Resistance

Risk resistance means that when risks occur, the system can fully mobilize its resources to maintain the internal stability of the system, including the stability and redundancy of the system.

Among them, stability is the ability of the system to maintain a safe state from the impact to the greatest extent, and it is the basis of system resilience. For high-speed railway construction, safety awareness, professional skills, and physical and mental status of construction personnel will directly affect their ability to deal with accidents and work quality; mechanical equipment as the basis for the normal operation of the system, its performance and state, material supply and quality are indispensable factors in construction. Standard construction is the premise of successful construction and personnel safety; the stability of environment system is ensured by analyzing the natural environment and working environment of the construction site and evaluating the risk considering the characteristics of goaf.

Redundancy is the ability to maintain the normal operation of the system or use other redundant measures in the construction process when the original equipment or machinery is damaged or the system suffers great losses [10]. In the high-speed railway subgrade construction system in goaf sites, in addition to reserving the necessary facilities and emergency facilities, setting subgrade deformation monitoring can assist in timely obtaining the data of subgrade structure stress changes, which is not only conducive to the timely modification of the design and construction scheme but also can predict accidents and dangers in advance, to take timely measures to ensure the safety of high-speed railway construction.

### 2.2.2. Recovery Capability

Recovery capability refers to the ability of the system to respond to adverse conditions through its organization capacity when the system is damaged resulting in structural damage or abnormal operation, to restore normal production as soon as possible, including the efficiency and adaptability of the system.

Efficiency is the responsibility of the system to take corresponding measures in the face of emergencies and the rapid recovery ability to restore the system to normal operation. For this system, the improvement of the hierarchical emergency management mechanism, the emergency rescue ability of personnel in the face of disasters, the location and radius of emergency channels and shelters, and the organizational level of personnel evacuation have an impact on the efficiency of the construction system.

Adaptability refers to the ability of personnel, material technology, and management to resist risks and learn afterward.

### 2.2.3. Resilience Evaluation Indexes of High-Speed Railway Subgrade Construction Systems in Goaf Sites

According to the above research, from the perspective of connotation, the four characteristic elements of resilience: stability, redundancy, efficiency, and adaptability are determined. Combined with the organization member system, material technology system, management system, and environmental system, the bibliometric method was also used,

searching for the keywords of "Resilience Evaluation", "High-speed Railway Engineering Resilience ", "Construction Resilience", and" Resilience Theory" in the China National Knowledge Infrastructure, the search year was set to 2000–2022, and 879 articles were retrieved. These articles were screened one by one, and on the basis of high frequency index statistics, the first level indicators were established from four aspects of stability, redundancy, efficiency and adaptability. Based on the connotation differences of these four first-level indicators, the second-level indicators were selected, and the resilience evaluation index system of high-speed railway subgrade construction systems in goaf sites was preliminarily established as shown in Table 1.

**Table 1.** Resilience evaluation indexes of high-speed railway subgrade construction systems in goaf sites.

| | Organization Member System | Material Technology System | Management System | Environment System |
|---|---|---|---|---|
| **Stability B1** | Safety cognition ability of personnel C1 [10,16] Professional skills of personnel C2 [10] Physical and mental state of personnel C3 [10] | Quality of material C4 [15,55] Supply and quality of Material C5 [15,55] Status and performance of mechanical equipment C6 [16] Construction specification C7 Construction technique C8 | Rules and regulations C9 [12] | Environmental risk assessment and countermeasures C10 [12] Surrounding environment and working environment C11 [56] |
| **Redundancy B2** | | Redundancy of facilities equipment C12 Redundancy of emergency facilities and material reserves C13 Monitoring system of subgrade deformation C14 | Emergency and safeguarding of accidents C15 | |
| **Efficiency B3** | Emergency rescue ability of personnel C16 | Ability to deal with environmental emergencies C17 [12] | Emergency management mechanism C18 [14] Emergency organization efficiency C19 [16] Emergency program C20 [15,55] | Emergency corridors and shelters C21 |
| **AdaptabilityB4** | Emergency response drill C22 [16] Safety education and training of personnel C23 [12] | Emergency apparatus C24 [15] | | Characteristics of goaf C25 |

## 3. ANP-Entropy Weight-Fuzzy Comprehensive Resilience Evaluation Model

### 3.1. ANP-Entropy Weight-Fuzzy Comprehensive Evaluation Theory

The research methods used in this study include the ANP, entropy weight method and the fuzzy comprehensive method. In the construction of high-speed railway subgrade in goaf sites, the resilience of the system is affected by many factors, and the factors are interdependent and the relationship is fuzzy. It is difficult to determine the influence degree of each index on the construction system by a single evaluation method. Therefore, the ANP-entropy weight-fuzzy method is used to evaluate the resilience. Firstly, the ANP method solves the problem of the interdependence of resilience indexes in the system, but the determined weight is greatly affected by subjectivity. The entropy weight method [57] is an objective weighting method, when it is used, the weight of each index is determined according to the known evaluation objects including indicators, which can reduce the sub-

jectivity of each index weighting. The ANP-entropy weight method [58] can fully consider the correlation among various evaluation factors, and correct the evaluation weight from experts based on the objective weight of entropy weight, which can eliminate the subjective error and make the evaluation results more reasonable. Then, the fuzzy comprehensive evaluation method [59] is based on the fuzzy mathematics theory. Considering the fuzziness of the evaluation target risk factors, it calculates and analyzes the indexes that are difficult to quantify through mathematical methods, including the safety cognitive ability of personnel in the construction system, professional skills, rules and regulations and other indexes. Compared with other evaluation models, it is more practical.

### 3.2. Construction of Resilience Evaluation Model for High-Speed Railway Subgrade Construction Systems in Goaf Sites

#### 3.2.1. Construct Factor Sets

It is assumed that the target layer is the resilience of the high-speed railway subgrade construction system in the mined-out area, $m$ first-level indexes $U = \{U_1, U_2, \ldots, U_m\}$, where $U_i$ represents the $i$th index in the first-level indexes. $n$ second-level indexes $U_i = \{u_{i1}, u_{i2}, \ldots, u_{in}\}$, where $u_{ij}$ denotes the $j$th second-level index in the $j$th first-level index.

#### 3.2.2. Establishment of Evaluation Sets

Assuming that the evaluation results are divided into several levels, we can establish the following evaluation sets $X$:

$$X = \{X_1, X_2, \ldots, X_n\} =$$
$$\{\text{very Low resilience, low resilience, medium resilience, high resilience, very high resilience}\}$$

#### 3.2.3. Establishment of Evaluation Matrix R

$$R = \begin{Bmatrix} r_{11} & r_{12} & \cdots & r_{1n} \\ r_{21} & r_{22} & \cdots & r_{2n} \\ \vdots & \vdots & \vdots & \vdots \\ r_{m1} & r_{m2} & \cdots & r_{mn} \end{Bmatrix} \tag{1}$$

In the formula, the index $U_i$ is rated as the membership degree of $X_j$ grade $r_{ij} = \frac{x}{y}$, $x$ means the number of people who are rated as $X_j$ grade by the evaluation index $u_{ij}$, and $y$ is the total number of people involved in the evaluation.

#### 3.2.4. Weight Calculation

(1) ANP Method to determine the subjective weight of the index $\omega_A$.

ANP is selected to calculate the subjective weight of each index in the resilience evaluation index system. Through Super Decision software, the subjective weight $\omega_A$ of the index is finally obtained.

(2) Determination of objective index weight $\omega_j$ by entropy weight method.

The calculation formula for determining the objective weight of evaluation index by the entropy weight method is as follows:

$$\begin{cases} X_{ij} = \frac{x_{ij} - m_j}{M_j - m_j} \\ p_{ij} = \frac{x_{ij}}{\sum_{i=1}^{n} x_{ij}} \\ e_j = -\frac{1}{\ln n} \sum_{i=1}^{n} p_{ij} \ln(p_{ij}) \end{cases} \tag{2}$$

$$\omega_j = \frac{1 - e_j}{\sum_{i=1}^{m} 1 - e_j} \tag{3}$$

In the formula, $n$ is the number of items to be evaluated; $i$ is the order number of evaluation items ($i = 1,2,3,\cdots,n$); $j$ is the order number of evaluation index; $p_{ij}$ is the weight of the $j$th second-level index of the $i$th first-level index; $e_j$ is the information entropy value of the $j$th evaluation index; $\omega_j$ is the weight of the $j$th evaluation index.

(3) Determination of comprehensive weight of index by ANP–entropy weight method.

$$\omega = \theta\omega_j + (1-\theta)\omega_A \tag{4}$$

In the above expression, $0 \leq \theta \leq 1$, we take $\theta = 0.5$.

### 3.2.5. Fuzzy Comprehensive Evaluation

According to the factor weight vector $\omega$ and fuzzy matrix $R$, the evaluation vector $B$ is obtained, namely:

$$B = \omega^\circ R \tag{5}$$

The evaluation grade can be obtained according to the maximum membership degree principle.

## 4. Case Application

### 4.1. Background of the Project

The study of resilience evaluation in this paper is conducted based on the DK259 + 135.95 − DK259 + 710.00 section of Taiyuan-Jiaozuo Railway [60] to verify the above research. It also assists to advance our understanding of the connotation of resilience, and even facilitates an accurate comprehension of construction in goaf sites on the resilience of high-speed railway systems. As shown in Figure 3, the proposed high-speed railway line passes through the No. 2 mined-out area, adjacent to three mined-out areas.

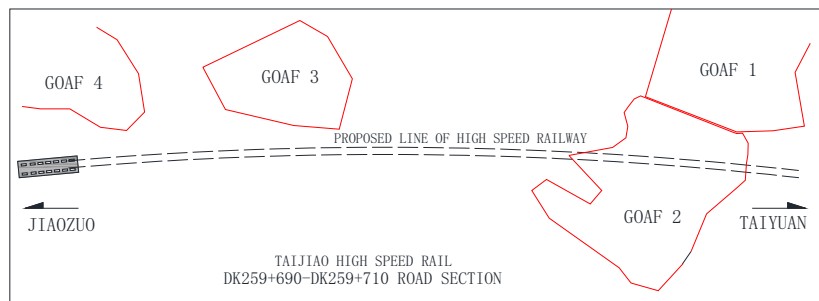

**Figure 3.** Schematic diagram of Taijiao high-speed railway route.

(1)  Characteristics of mined-out areas

The rock mass structure of the goaf below this section is relatively complete, and there are few adverse geological structures (Figure 4); the depth of the goaf is 45.2 m–60.3 m, and the mining thickness is 1.2 m–6.5 m. In the preliminary investigation, it is found that most of the goaf areas have unstable roofs.

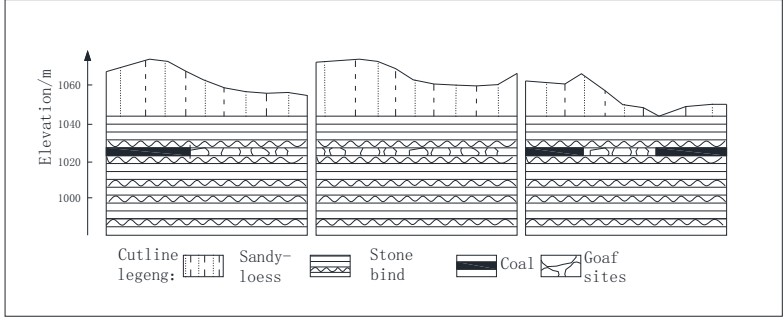

**Figure 4.** Geological section of goaf sites on the Taijiao railway.

(2)    Project management and emergency disposal

In project management, a project manager responsibility system is established, and a management system is implemented with the project manager as the core, with a clear division of labor, clear responsibility to everyone and hierarchical management. In emergency disposal, the emergency rescue plan for safety accidents in engineering construction is established, which must meet the characteristics of the project and prevent great damage from natural disasters and human accidents, and an emergency management mechanism is built to improve the efficiency of emergency organization.

(3)    Staffing and mechanical equipment

According to the relevant requirements of the Safety Production Law, the policy of "safety first, prevention first, comprehensive management" is implemented and the education and training of safety products and safety operation skills are carried out. According to the principle of mechanized operation and industrial production, the selection of main construction machinery and equipment should be able to meet the requirements of construction and ensure the construction capacity and quality requirements of the project.

(4)    Hydrogeological and climatic conditions

Groundwater along the Taijiao railway is mainly quaternary pore phreatic water, recharged by atmospheric precipitation and groundwater recharge, with a groundwater depth of 9.8 m–22 m. Rainfall is mainly concentrated in July and August, accounting for 80% of the whole year, which belongs to the warm climate zone. The construction scheme was prepared according to the rainy season, the required materials were prepared in advance, and the drainage plan of the construction site was made according to the terrain of the site.

According to the resilience indexes of the system selected above, 14 experts and scholars in related fields were invited (among them, four experts from the field of management science and engineering, and six experts in the field of civil engineering, four technical engineers in the field of high-speed railways). Based on the actual situation of Taijiao railway subgrade construction, the importance and influence of the system were scored according to the resilience indexes. After collecting questionnaire responses, SPSS was used to test the reliability of the questionnaire responses. The results show that the overall Cronbach's alpha coefficient of the questionnaire is greater than 0.8, indicating that the overall reliability is good.

*4.2. Determination of Index Weight*

Firstly, the 14 experts were invited to fill in an analytical network process (ANP) questionnaire indicating the relative importance of each indicator, and when all questionnaires were collected, the judgment matrix was constructed. Super Decisions was used to test the inconsistency and calculate the weight value of each indicator (Table 2).

Secondly, the entropy weight method is used to determine the objective weight of the evaluation index according to Equations (2) and (3), and the normalization is carried out. The results are shown in Table 3.

Finally, the comprehensive weight of the evaluation index is calculated according to Equation (4), and the results are shown in Table 4.

**Table 2.** Subjective weights and ranking of resilience indexes.

| Target | First Grade Indexes | Weight | Second Grade Indexes | Weights | Subjective Weights | Sort |
|---|---|---|---|---|---|---|
| *A* Resilience of High-Speed Railway Subgrade Construction System in Mined-Out Area | Stability B1 | 0.4559 | Safety cognition ability of personnel C1 | 0.0227 | 0.0057 | 21 |
| | | | Professional skills of personnel C2 | 0.1077 | 0.0269 | 15 |
| | | | Physical and mental state of personnel C3 | 0.0118 | 0.0030 | 24 |
| | | | Quality of material C4 | 0.0121 | 0.0030 | 23 |
| | | | Supply of material C5 | 0.0221 | 0.0055 | 22 |
| | | | Status and performance of mechanical equipment C6 | 0.0438 | 0.0109 | 20 |
| | | | Construction specification C7 | 0.0498 | 0.0124 | 18 |
| | | | Construction technique C8 | 0.1176 | 0.0294 | 13 |
| | | | Rules and regulations C9 | 0.1197 | 0.0299 | 12 |
| | | | Environmental risk assessment and countermeasures C10 | 0.3929 | 0.0981 | 2 |
| | | | Surrounding environment and working environment C11 | 0.0998 | 0.0249 | 17 |
| | Redundancy B2 | 0.1301 | Redundancy of facility equipment C12 | 0.0084 | 0.0011 | 25 |
| | | | Redundancy of emergency facilities and material reserves C13 | 0.2099 | 0.0273 | 14 |
| | | | Monitoring system of subgrade deformation C14 | 0.4603 | 0.0598 | 8 |
| | | | Emergency and safeguarding of accidents C15 | 0.3214 | 0.0418 | 11 |
| | Efficiency B3 | 0.1849 | Emergency rescue ability of personnel C16 | 0.0372 | 0.0113 | 19 |
| | | | Ability to deal with environmental emergencies C17 | 0.3859 | 0.1170 | 1 |
| | | | Emergency management mechanism C18 | 0.1989 | 0.0603 | 7 |
| | | | Emergency organization efficiency C19 | 0.1449 | 0.0439 | 10 |
| | | | Emergency program C20 | 0.1471 | 0.0446 | 9 |
| | | | Emergency corridors and shelters C21 | 0.0861 | 0.0261 | 16 |
| | Adaptability B4 | 0.2291 | Emergency response drill C22 | 0.2601 | 0.0825 | 4 |
| | | | Safety education and training of personnel C23 | 0.2854 | 0.0905 | 3 |
| | | | Emergency apparatus C24 | 0.1988 | 0.0630 | 6 |
| | | | Characteristics of goaf C25 | 0.2557 | 0.0811 | 5 |

**Table 3.** Objective weights and ranking of resilience indexes.

| First Grade Indexes | Weight | Second Grade Indexes | Objective Weights | Sort |
|---|---|---|---|---|
| Stability B1 | 0.4412 | Safety cognition ability of personnel C1 | 0.0383 | 15 |
| | | Professional skills of personnel C2 | 0.0478 | 1 |
| | | Physical and mental state of personnel C3 | 0.0376 | 20 |
| | | Quality of material C4 | 0.0401 | 10 |
| | | Supply of material C5 | 0.0459 | 3 |
| | | Status and performance of mechanical equipment C6 | 0.0378 | 18 |
| | | Construction specification C7 | 0.0368 | 21 |
| | | Construction technique C8 | 0.0398 | 11 |
| | | Rules and regulations C9 | 0.0408 | 8 |
| | | Environmental risk assessment and countermeasures C10 | 0.0417 | 7 |
| | | Surrounding environment and working environment C11 | 0.0346 | 24 |
| Redundancy B2 | 0.1719 | Redundancy of facility equipment C12 | 0.0452 | 4 |
| | | Redundancy of emergency facilities and material reserves C13 | 0.0467 | 2 |
| | | The monitoring system of subgrade deformation C14 | 0.0417 | 6 |
| | | Emergency and safeguarding of accidents C15 | 0.0383 | 16 |
| Efficiency B3 | 0.2298 | Emergency rescue ability of personnel C16 | 0.0382 | 17 |
| | | Ability to deal with environmental emergencies C17 | 0.0364 | 23 |
| | | Emergency management mechanism C18 | 0.0387 | 14 |
| | | Emergency organization efficiency C19 | 0.0402 | 9 |
| | | Emergency program C20 | 0.0395 | 13 |
| | | Emergency corridors and shelters C21 | 0.0368 | 22 |
| Adaptability B4 | 0.1569 | Emergency response drill C22 | 0.0346 | 25 |
| | | Safety education and training of personnel C23 | 0.0377 | 19 |
| | | Emergency apparatus C24 | 0.0451 | 5 |
| | | Characteristics of goaf C25 | 0.0395 | 12 |

**Table 4.** Comprehensive weights and ranking of resilience indexes.

| First Grade Indexes | Weight | Second Grade Indexes | Comprehensive Weights | Sort |
|---|---|---|---|---|
| Stability B1 | 0.3456 | Safety cognition ability of personnel C1 | 0.0220 | 23 |
| | | Professional skills of personnel C2 | 0.0373 | 12 |
| | | Physical and mental state of personnel C3 | 0.0203 | 25 |
| | | Quality of material C4 | 0.0216 | 24 |
| | | Supply of material C5 | 0.0257 | 18 |
| | | Status and performance of mechanical equipment C6 | 0.0244 | 21 |
| | | Construction specification C7 | 0.0246 | 20 |
| | | Construction technique C8 | 0.0346 | 15 |
| | | Rules and regulations C9 | 0.0354 | 14 |
| | | Environmental risk assessment and countermeasures C10 | 0.0699 | 2 |
| | | Surrounding environment and working environment C11 | 0.0298 | 17 |
| Redundancy B2 | 0.1509 | Redundancy of facility equipment C12 | 0.0231 | 22 |
| | | Redundancy of emergency facilities and material reserves C13 | 0.0370 | 13 |
| | | The monitoring system of subgrade deformation C14 | 0.0508 | 7 |
| | | Emergency and safeguarding of accidents C15 | 0.0400 | 11 |
| Efficiency B3 | 0.2665 | Emergency rescue ability of personnel C16 | 0.0247 | 19 |
| | | Ability to deal with environmental emergencies C17 | 0.0767 | 1 |
| | | Emergency management mechanism C18 | 0.0495 | 8 |
| | | Emergency organization efficiency C19 | 0.0421 | 9 |
| | | Emergency program C20 | 0.0420 | 10 |
| | | Emergency corridors and shelters C21 | 0.0315 | 16 |
| Adaptability B4 | 0.237 | Emergency response drill C22 | 0.0585 | 5 |
| | | Safety education and training of personnel C23 | 0.0641 | 3 |
| | | Emergency apparatus C24 | 0.0541 | 6 |
| | | Characteristics of goaf C25 | 0.0603 | 4 |

### 4.3. Fuzzy Comprehensive Evaluation Based on Comprehensive Weights

The resilience grade evaluation set $X$ of the high-speed railway subgrade construction system in the goaf area is described as Table 5, and the questionnaire data of 14 experts were analyzed. The evaluation matrix of second-grade indexes is as follows:

$$
R_1 = \begin{Bmatrix} 0.21 & 0 & 0.29 & 0.14 & 0.36 \\ 0.07 & 0.07 & 0.29 & 0.29 & 0.28 \\ 0.07 & 0.07 & 0.36 & 0.36 & 0.14 \\ 0 & 0 & 0.36 & 0.36 & 0.28 \\ 0 & 0.07 & 0.28 & 0.42 & 0.23 \\ 0 & 0.07 & 0.43 & 0.21 & 0.29 \\ 0.07 & 0.07 & 0.43 & 0.29 & 0.21 \\ 0 & 0 & 0.14 & 0.71 & 0.15 \\ 0.07 & 0.14 & 0.36 & 0.36 & 0.17 \\ 0.07 & 0.14 & 0.36 & 0.36 & 0.17 \\ 0 & 0.14 & 0.5 & 0.36 & 0 \end{Bmatrix}
R_2 = \begin{Bmatrix} 0.07 & 0.14 & 0.5 & 0.14 & 0.15 \\ 0.07 & 0.07 & 0.36 & 0.43 & 0.07 \\ 0 & 0.07 & 0.29 & 0.29 & 0.35 \\ 0 & 0.21 & 0.29 & 0.29 & 0.21 \end{Bmatrix}
$$

$$
R_3 = \begin{Bmatrix} 0.07 & 0.14 & 0.07 & 0.29 & 0.43 \\ 0 & 0.07 & 0.57 & 0.21 & 0.15 \\ 0.14 & 0.07 & 0.36 & 0.36 & 0.07 \\ 0.07 & 0 & 0.36 & 0.36 & 0.07 \\ 0 & 0.21 & 0.28 & 0.43 & 0.08 \\ 0 & 0 & 0.57 & 0.29 & 0.14 \end{Bmatrix}
R_4 = \begin{Bmatrix} 0.29 & 0.07 & 0.21 & 0.36 & 0.07 \\ 0.14 & 0.15 & 0.21 & 0.5 & 0 \\ 0 & 0.21 & 0.28 & 0.43 & 0.08 \\ 0 & 0 & 0.57 & 0.29 & 0.14 \end{Bmatrix}
$$

The evaluation set $B_i$ of second grade indexes is:

$$
B_1 = (0.0592, 0.0635, 0.1880, 0.5013, 0.1880)
$$
$$
B_2 = (0.0442, 0.1435, 0.2430, 0.2655, 0.3038)
$$
$$
B_3 = (0.0800, 0.0620, 0.4959, 0.2382, 0.1240)
$$
$$
B_4 = (0.1616, 0.1121, 0.3331, 0.3099, 0.0833)
$$

From formula (5), the comprehensive evaluation is:

$$
B = \omega^\circ R = (0.0370, 0.0289, 0.0917, 0.2285, 0.0857)
$$

According to the principle of maximum membership, the resilience grade of the region is "high resilience".

**Table 5.** Resilience grade of high-speed railway subgrade construction systems in goaf sites.

| | Grade | Grade Description |
|---|---|---|
| 1 | Very Low Resilience | Under the influence of goaf, the system has poor resistance ability and absorption ability to the possible risks in the process of subgrade construction. After the risk impacts, the ability of recovery and adaptability of the system is insufficient. It will take a long time for the system to recover from the impacts. |
| 2 | Low Resilience | Under the influence of goaf, the system has poor resistance ability and absorption ability to the possible risks during subgrade construction, poor ability of recovery and adaptability after risk impacts, and it will take a certain time for the system to recover from the impacts. |
| 3 | Middle Resilience | Under the influence of goaf, the system has poor resistance ability and absorption ability to the possible risks in the process of subgrade construction. After the risk impacts, the system has a good ability of recovery and adaptability, and the system can recover from the impacts in a certain time. |

**Table 5.** *Cont.*

| | Grade | Grade Description |
|---|---|---|
| 4 | High Resilience | Under the influence of goaf, the system has better resistance ability and absorption ability to the possible risks in the process of subgrade construction and has a better ability of recovery and adaptability after the risk impacts. The system can return to normal safety state from the impacts in a certain time. |
| 5 | Very High Resilience | Under the influence of goaf, the system has good resistance ability and absorption ability to the possible risks in the process of subgrade construction, and has a good ability of recovery and adaptability after the risk impacts. The system can recover from the impacts in a certain period of time. |

*4.4. Weight Analysis of Resilience Index*

By analyzing and sorting the calculated weight of each resilience index, it is found that the two factors that have the greatest impact on the resilience of the system in the first-level index are stability and efficiency. Thus, for the subgrade construction system, the most important is stability, which is the foundation of high-speed railway subgrade construction system resilience. System stability should be the main management objective in construction and environmental risk assessment and response capacity should be improved to ensure the level of resistance in the event of disasters; the second important factor is efficiency, through improving the hierarchical emergency management mechanism and the emergency rescue ability of personnel in the face of disasters, the responsibility of the system to take corresponding measures in the face of emergencies is strengthened to ensure the system can recover as soon as possible. The weight analysis of the second-level resilience indexes shows that the coping ability of environmental emergencies, environmental risk assessment and response measures have a great impact on the resilience of the system. Therefore, the resilience evaluation of this research system is inseparable from the influence of the surrounding construction environment. Attention should be paid to the activation mechanism of the goaf foundation and the deformation mechanism of the subgrade during construction. Combined with the actual situation of the project, targeted measures are taken to improve the resilience of the system and improve the risk resistance and recovery ability of the high-speed railway subgrade construction system in the mined-out area.

Environmental impact analysis was constructed based on the characteristics of goaf in practical engineering, and it was found that the construction of the Taiyuan-Jiaozuo intercity railway inevitably affects the environment on both sides of the line, effective measures were put forward in the construction and operation of the project [61], which are consistent with the model evaluation results, indicating that the evaluation results have certain credibility. Finally, our research results are verified by the Taijiao Railway environmental impact report results [61] and the actual situation of the questionnaire survey: to start with, the environmental impact report shows that the impact of the environment on the resilience of the construction system is very important, which is consistent with the resilience evaluation results that environmental factors account for a large part of the overall factors. In addition, based on the fuzzy comprehensive evaluation of comprehensive weight, we obtained the following data through a questionnaire survey of 14 experts, and the results were analyzed as shown in Table 6. C2, C10, C13–15, C17–20 and C22–25 have a very high resilience evaluation grade and account for 50% of the overall index. C1–2, C4, C5, C7–15 and C17–25 have a grade evaluation above medium resilience, accounting for 88% of the overall index. Only C3, C6 and C16 have low resilience, which indicates that the overall resilience level of this area is high, and the results are consistent with the resilience evaluation grade of the ANP-entropy-fuzzy comprehensive evaluation model.

**Table 6.** List of comprehensive evaluation.

| Second Grade indexes | Very High Resilience (100–90) | High Resilience (89–80) | Middle Resilience (79–70) | Low Resilience (69–60) | Very Low Resilience (59–0) |
|---|---|---|---|---|---|
| Safety cognition ability of personnel C1 | | 12 | 2 | | |
| Professional skills of personnel C2 | 2 | 10 | 2 | | |
| Physical and mental state of personnel C3 | | 9 | 4 | 2 | |
| Quality of material C4 | | 10 | 4 | | |
| Supply of material C5 | | 9 | 5 | | |
| Status and performance of mechanical equipment C6 | | 8 | 5 | 1 | |
| Construction specification C7 | 1 | 10 | 3 | | |
| Construction technique C8 | | 9 | 5 | | |
| Rules and regulations C9 | 1 | 9 | 4 | | |
| Environmental risk assessment and countermeasures C10 | 2 | 11 | 1 | | |
| Surrounding environment and working environment C11 | | 8 | 6 | | |
| Redundancy of facility equipment C12 | | 11 | 3 | | |
| Redundancy of emergency facilities and material reserves C13 | 2 | 10 | 2 | | |
| The monitoring system of subgrade deformation C14 | 4 | 8 | 2 | | |
| Emergency and safeguarding of accidents C15 | 3 | 8 | 3 | | |
| Emergency rescue ability of personnel C16 | | 10 | 3 | 1 | |
| Ability to deal with environmental emergencies C17 | 2 | 12 | | | |
| Emergency management mechanism C18 | 1 | 7 | 6 | | |
| Emergency organization efficiency C19 | 3 | 8 | 3 | | |
| Emergency program C20 | 3 | 6 | 5 | | |
| Emergency corridors and shelters C21 | | 5 | 9 | | |
| Emergency response drill C22 | 2 | 8 | 4 | | |
| Safety education and training of personnel C23 | | 9 | 5 | | |
| Emergency apparatus C24 | 3 | 10 | 1 | | |
| Characteristics of goaf C25 | | 10 | 4 | | |

## 5. Discussion

(1) Improving the engineering resilience is critical to safeguard the high reliability, low disaster consequences and ensure rapid recovery of the project from the perspective of sustainability. The recovery ability and reliability ability are the main determinants of engineering resilience, only by paying attention to the control of the two abilities at the same time can the target resilience value be achieved in the optimal way.

(2) The restoring capacity of engineering structures is crucial to the improvement of engineering resilience. However, it is difficult to be repaired when the engineering structure is damaged when the goaf is deeply buried in the underground, and in the existing research, there are few studies on the recoverability of goaf sites and the rapid repair technology of damaged structures, and the cost of methods to improve the recoverability is relatively high. Therefore, further exploration of the engineering resilience is worthwhile to economically and effectively improve the rapid recovery ability of the high-speed railway subgrade construction systems in goaf after natural disasters or human destruction.

(3) The resilience evaluation index and evaluation method are the basis of resilience evaluation, whereas the uncertainty and variability of the system are increased because of the action of goaf. For example, in the case of groundwater, the continuous failure process caused by water-soil coupling is complex, and the indicators of system performance are complex and diverse. Moreover, the present studies did not conduce the resilience evaluation indexes that can comprehensively consider the stability, redundancy, efficiency,

adaptability and sustainability of the high-speed railway construction systems in goaf sites. Therefore, it is urgent to explore reasonable and convenient resilience evaluation indexes and evaluation methods in the next step.

(4) At present, in the field of geotechnical engineering, robustness analysis has been proposed based on traditional reliability theory at home and abroad [62–65]. The robustness means that the control system maintains some other performance characteristics under certain parameter perturbation [66]. From the above analysis, in the construction system of high-speed railway subgrade in mined-out areas, the geotechnical and underground structures in the mined-out area have great uncertainty and variability. For the sensitivity of these uncertain and variable parameters, the resilience evaluation system can deepen the traditional reliability theory by robustness analysis. The robustness can be considered in engineering design or specification formulation in the future so it should be further developed.

(5) In practical applications, resilience should focus on the combination of evaluation theory and practical research, study the key elements of resilience and their relationships, pay attention to the influence of human activities, and reflect its guiding significance for development planning. The future resilience research should start from the system as a whole, and build a dynamic resilience evaluation, monitoring and prediction system to promote resilience evaluation systematization, standardization and normalization to meet the needs of national and regional sustainability.

## 6. Conclusions

The present work aimed to assess the resilience of high-speed railway subgrade construction systems above goaf. The following conclusions can be obtained through the above analysis:

(1) The introduction of resilience theory provides new ideas and methods for safety management of high-speed railway construction sites above goaf. Combined with the system composition and characteristic elements, the evaluation index system of high-speed railway subgrade construction system resilience in goaf sites was established, which includes four first-level indexes and 25 second-level indexes, covering the four elements of labor, machinery, environment and management and different dimensions of system resilience in construction, meaning that the result has a high credibility.

(2) We used the ANP-entropy weight-fuzzy evaluation model to evaluate resilience. This model not only considers the possible interrelationship among various resilience indexes, reduces the possibility that the weight calculation is greatly affected by the subjective and is thus inaccurate, but also takes into account the fuzziness and randomness of the index boundaries, therefore, the scientificity of the evaluation process and evaluation results are ensured.

(3) The resilience grade of the Taijiao high-speed railway subgrade construction was evaluated, and the result is "high resilience", indicating that under the influence of goaf, the system can return to a normal safety state from the impacts within a certain time. Through the determination of resilience grade, the indicators' weight calculation and the introduction of optimization measures were carried out. Finally, the research results provide new safety management ideas for high-speed railway projects constructed in mined-out areas from the perspective of sustainability, and optimize the results-oriented safety risk management mode, which is beneficial for improving the safety resilience and risk resistance level of the construction site.

**Author Contributions:** H.W.: writing—review and editing; J.Z.: writing—original draft preparation; Z.D. (Zhiyuan Dun): data curation; J.C.: investigation; H.L.: writing—review and editing; Z.D. (Zhilin Dun): investigation. All authors have read and agreed to the published version of the manuscript.

**Funding:** This research was funded by (1) the National Natural Science Foundation of China, grant number U1810203, and (2) General Research Project of Humanities and Social Sciences in Colleges and Universities of Henan Province, grant number 2023-ZDJH-145.

**Institutional Review Board Statement:** Not applicable.

**Informed Consent Statement:** Not applicable.

**Data Availability Statement:** Not applicable.

**Acknowledgments:** The writers wish to acknowledge the financial support to this research by the National Fund Committee. The kind assistance and valuable contributions of the staff of the School of Civil Engineering, Henan Polytechnic University who provided field investigation and field testing, are gratefully acknowledged.

**Conflicts of Interest:** All authors declare that they have no conflict of interest.

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
