# Peer review of "Resilience Evaluation of High-Speed Railway Subgrade Construction Systems in Goaf Sites"

_sustainability, doi:10.3390/su14137806_

Round 1
Reviewer 1 Report
1. The title is suggested to change to “Resilience Evaluation of High-speed Railway Subgrade Construction System in Goaf Sites”.
2. The research scope of the study in China is somehow insufficient. It is suggested to review or analyze some case studies of high-speed railway operation accidents in goaf sites for the other countries.
3. The “coupled chain disaster of goaf” is not reflected in Fig.1, please revise it.
4. It is suggested to provide a schematic diagram of the goaf for the project example.
5. What is the difference between the weights and subjective weights summarized in Table 3?
6. Why there are no primary indexes and their weights in Tables 4 and 5? Please explain.
7. There is no reliable verification for the evaluation results of project example.
8. What are the differences and connections between resilience assessment and risk safety management? Can the differences and connections among both of them be illustrated?
9. The last sentence in the Abstract should be replaced to the end of the Conclusions.
10. The English expression should be further polished to avoid the grammatical errors or confusions.
11. Scientific issues need to be further summarized in the Introduction.
12. The format of references in Table 2 is not standard and it is suggested to be further standardized.
Author Response
Dear reviewer,
Thanks for your letter and for reviewers’ comments concerning our manuscript entitled” Resilience Evaluation of High-speed Railway Construction System in Goaf Sites “(Manuscript ID: sustainability-1765837). Those comments are all valuable and helpful for revising and improving our paper. We have studied all comments carefully and have made conscientious corrections. I believe that the addressing of these comments has greatly improved the quality of this manuscript.
Responses to reviewer’ comments:
Question1:The title is suggested to change to “Resilience Evaluation of High-speed Railway Subgrade Construction System in Goaf Sites”.
Answer: Thank you for your comments. It is true that the word "subgrade" should be added to the title to emphasize the role of high speed railway subgrade in the study, which has been revised in the title section.
Question2:The research scope of the study in China is somehow insufficient. It is suggested to review or analyze some case studies of high-speed railway operation accidents in goaf sites for the other countries.
Answer: Thank you for your comments. We have searched for relevant reports and studies, but there are few accident cases of high-speed rail passing through goaf in foreign countries, so we added one foreign cases: No. 70 High-speed railway in the Midwest of the United States passes through the goaf sites, the foundation activation in the goaf leads to the settlement and uneven deformation of railway roadbed, but we will continue to look for relevant cases in the following research, and we hope to get your useful guidance.(Paragraph 1,page 2)
Question3:The “coupled chain disaster of goaf” is not reflected in Fig.1, please revise it.
Answer:We thank the reviewer for this comment. Indeed, the characteristics of coupled chain disasters of goaf are not reflected in Figure 1, so we modify Figure 1 and give a more detailed description of coupled chain disasters in the line 3-7 of the first paragraph of the introduction. (Page2)
Question4:It is suggested to provide a schematic diagram of the goaf for the project example.
Answer:We are grateful for your kind comment. According to your suggestion, we added the map of geological section of goaf sites of Taijiao railway in 4.1” Background of the Project” (Page 13)
Question5:What is the difference between the weights and subjective weights summarized in Table 3?
Answer: The ' weight ' in Table 3(New Table 2) refers to the ' normalized by cluster ' calculated in the super decision software, and the ' subjective weight ' in the table refers to the ' limit ' calculated. Because “the normalized by cluster” is a local matrix weight without normalization, it should be used from the global perspective.
Question 6: Why there are no primary indexes and their weights in Tables 4 and 5? Please explain.
Answer: We thank the reviewer for raising this question. Because the weight of the first-level indicators in Table 4(New Table 3) and Table 5(New Table 4) is calculated by adding the second-level indicators, so I did not write it in the table before, but this is not standardized, it is honoured that you pointed out this point, I have modified this part in new Table 3 and new Table 4, and the corresponding part of the later modified.
Question 7:There is no reliable verification for the evaluation results of project example.
Answer: Your suggestion is very useful, we really did not introduce this part in detail in the previous manuscript. Our research results are verified by the Taijiao Railway environmental impact report results and the actual situation of the questionnaire survey: First, the environmental impact report shows that the impact of the environment on the resilience of the construction system is very important, which is consistent with the resilience evaluation results that environmental factors account for a large part of the overall factors; in addition, based on the fuzzy comprehensive evaluation of comprehensive weight, we obtained the following data through a questionnaire survey of 14 experts, and the results were analyzed as shown in the following table. Firstly, C2, C10, C13-15, C17-20, and C22-25 have very high resilience evaluation grade and account for 50% of the overall index. C1-2, C4, C5, C7-15, C17-25 have grade evaluation above medium resilience, accounting for 88% of the overall index. Only C3, C6 and C16 have low resilience, which indicates that the overall resilience level of this area is high, and the results are consistent with the resilience evaluation grade of ANP-entropy-fuzzy comprehensive evaluation model. This part is also added in section “4.4. Weight Analysis of Resilience Index”.(Page17-18)
Second Grade indexes |
Very High Resilience(100-90) |
High Resilience(89-80) |
Middle Resilience(79-70) |
Low Resilience(69-60) |
Very Low Resilience(59-0) |
Safety cognition ability of personnelC1 |
|
12 |
2 |
|
|
Professional skills of personnelC2 |
2 |
10 |
2 |
|
|
Physical and mental state of personnelC3 |
|
9 |
4 |
2 |
|
Quality of materialC4 |
|
10 |
4 |
|
|
Supply of materialC5 |
|
9 |
5 |
|
|
Status and performance of mechanical equipmentC6 |
|
8 |
5 |
1 |
|
Construction specificationC7 |
1 |
10 |
3 |
|
|
Construction techniqueC8 |
|
9 |
5 |
|
|
Rules and regulationsC9 |
1 |
9 |
4 |
|
|
Environmental risk assessment and countermeasuresC10 |
2 |
11 |
1 |
|
|
Surrounding environment and working environmentC11 |
|
8 |
6 |
|
|
Redundancy of facility equipmentC12 |
|
11 |
3 |
|
|
Redundancy of emergency facilities and material reservesC13 |
2 |
10 |
2 |
|
|
The monitoring system of subgrade deformation C14 |
4 |
8 |
2 |
|
|
Emergency and safeguarding of accidentsC15 |
3 |
8 |
3 |
|
|
Emergency rescue ability of personnelC16 |
|
10 |
3 |
1 |
|
Ability to deal with environmental emergenciesC17 |
2 |
12 |
|
|
|
Emergency management mechanismC18 |
1 |
7 |
6 |
|
|
Emergency organization efficiencyC19 |
3 |
8 |
3 |
|
|
Emergency programC20 |
3 |
6 |
5 |
|
|
Emergency corridors and sheltersC21 |
|
5 |
9 |
|
|
Emergency response drillC22 |
2 |
8 |
4 |
|
|
Safety education and training of personnelC23 |
|
9 |
5 |
|
|
Emergency apparatusC24 |
3 |
10 |
1 |
|
|
Characteristics of goafC25 |
|
10 |
4 |
|
|
Question8:What are the differences and connections between resilience assessment and risk safety management? Can the differences and connections among both of them be illustrated?
Answer: We are so grateful for your kind question.
Firstly,differences between resilience assessment and risk safety management:
(1) From the perspective of conceptual connotation, risk is considered to be the product of the probability of a disaster and the result or expected loss of a disaster ; resilience refers to the control force that the system can withstand a series of changes and still maintain functions and structures,it also has the ability of self-organizing, establishing and promoting adaptive learning.
(2) From the process point of view: risk research emphasizes the probability of risk events and the degree of loss, ignoring the response of the system itself after the risk events; the resilience research emphasizes that when the system is affected by adverse events, it is prepared and strategic to take active measures to reduce the impact of adverse events, restore the original structure and function, and form a new model to resist risks.
Next is the relationship between them:
(1) From the target point of view: for a system, the ultimate goal of risk research and resilience research is to ensure system security;
(2) From the research perspective, in order to ensure the safety of the construction system, it is necessary to carry out risk management, but at the same time, there are deficiencies in risk management : risk is a forward-looking prior analysis, that is, to find the factors that threaten the system as much as possible, and on this basis to find reasonable countermeasures, while ignoring the resistance ability of the system when it is damaged ; risk adopts the logic of “ cause-result ”, stops at the destruction of the system by adverse events on the boundary of thinking, and does not pay enough attention to the subsequent events of the destruction of the system. Due to the limitations of risk, people try to change the research perspective: if the system is bound to bear the harm of unexpected events, in a prepared and strategic way to reduce the impact of adverse events on the system, when the harm comes, it can strive for a more favorable situation, and take corresponding measures to repair the damaged system, which is the research idea of " resilience."
Reference:
[1] Wang, Y.; Jin, H.; Fu, S. S.; W, X. Y.; Wu, B. Thinking transformation of system security: a comparative study of risk and resilience. China Saf. Sci. J. 2018, 28, 62-68.
[2] James, K. M. Natural Hazards: Explanation and Integrationby Graham A. Tobin; Burrell E. Montz. Econ. Geogr. 1999, 75, 102–104.
[3] Deyle, R.; French, S.; Olshansky, R. Hazard assessment: The factual basis for planning and mitigation. Cooperating with nature: Confronting natural hazards and land use planning for sustainable communities, 1998, 119~166.
[4] Wang, H. R.; Yang, Y. F.; Yang, R. X; Deng, C. Y.; Gong, S. X. The uncertainty of the system of water resources security thinking: from risk to toughness. J. north. Chin. Water conservancy and hydropower. (nat. sci. ed.), 2022 lancet (01): 1-8.
Question9:The last sentence in the Abstract should be replaced to the end of the Conclusions.
Answer: Thanks for your reminding. The abstract and conclusion have been revised.
Question10:The English expression should be further polished to avoid the grammatical errors or confusions.
Answer: Your suggestions are very important and useful to us. We have revised the mistakes in the article, but they may not be perfect. We hope you can kindly comment on them.
Question 11: Scientific issues need to be further summarized in the Introduction.
Answer: Thanks for the helpful comments. According to your suggestion, we expanded the introduction and established a comprehensive research framework to better describe the research questions and research ideas.
Question12: The format of references in Table 2 is not standard and it is suggested to be further standardized.
Answer: Thanks for the reviewer pointing out our mistake and providing us with helpful suggestions. This manuscript has been revised in accordance with the correct reference format, and it is worth noting that the original Table 2 has been deleted, and the reference references cited are marked out in Table 1.
We are looking forward to your reply.
Reviewer 2 Report
This manuscript is about the resilience evaluation of high-speed railway construction system in goaf sites.
The paper is well written and composed. Needs some improvement.
Abstract:
Add something about the quantitative results in the abstract.
Add something about the benefits of the research.
2. Resilience of High-speed Railway Subgrade Construction System in Goaf Sites
Please establish the already conducted research in the field. Please improve with the help of (a) Previously used Variables and Previously used modelling techniques for similar type of research. Add 30-40 new references and find gap in existing literature. [Major Issue-Please add details with latest references].
-Can you please establish the research gap [Please add]
Materials and Methods:
-A comprehensive research framework missing- to follow the research is steps are missing. Add framework-flowchart and write this section in stepwise pattern.
. This portion should also be written in step wise pattern so that readers can understand the procedure for implementation purpose. [Please add]
Author Response
Dear reviewer,
Thanks for your letter and comments concerning our manuscript entitled” Resilience Evaluation of High-speed Railway Subgrade Construction System in Goaf Sites “(Manuscript ID: sustainability-1765837). Those comments are all valuable and helpful for revising and improving our paper. We have studied all comments carefully and have made conscientious corrections. I believe that the addressing of these comments has greatly improved the quality of this manuscript.
We will reply to your comments in three aspects:
- Quantitative results description and the benefits of research should be added in the abstract; 2.eatablish the already conducted research in the field; 3.eatablish the comprehensive research framework.
Answer 1: Thank you very much for your careful review work and helpful suggestions. According to your opinion, we have revised the abstract and added the quantitative results description and the benefits of research. (Abstract, page1).
Answer 2: Thanks for your kindly and helpful suggestions. In the introduction, we added some references related to the establishment of resilience index system and resilience evaluation method. (Introduction, page 4)
Answer 3: We really appreciated the helpful suggestions. Indeed, by establishing a comprehensive research framework in our article, it may help readers better understand our thinking and process. Here's the comprehensive research framework we set up: (Figure 2, page5)
We are looking forward to your reply.

Reviewer 3 Report
1. Resilience evaluation is performed for a high-speed railway subgrade construction system in goaf site with Taijiao high-speed railway as an example. The characteristics of resilience are discussed from risk resistance and recovery capability based on the concept of sustainability. The characteristic elements of resilience, including stability, redundancy, efficiency, and adaptability, are discussed. The resilience influencing factor matrix is established based on the organization member system, material technology system, management system, and environmental system. Resilience index system and evaluation model are established. ANP-entropy weight-fuzzy method are introduced to evaluate the resilience. It is interesting and useful.
2. However, most of the manuscript is discussing the concept of resilience and a subjective evaluation system. The evaluation system is also personnel-dependent although both subjective weight and objective index are considered. It is not clear what is the core scientific problem of the manuscript. Is it the concept? resilience index system? or the evaluation system?
3. The abstract is too long and there is no description on the establishment method of first grade indexes and second grade indexes.
4. The introduction should be improved. Some references related to this article should be used.
5. What is new of this research? What is the major aim of the article?
6. In section 2.1, According to the previous research results, the construction of high-speed railway subgrade in goaf sites is divided into organization member system, material technology system, management system, and environmental system. Corresponding references should be clearly indicated.
7. The references in Table 2 are incorrectly formatted.
8. English writing is generally good; some tenses need further confirmation.
9. The origin of resilience indexes should be more clearly explained.
10. All formulas should be numbered.
11. There is an error in line 214.
12. The section number 1, 2, 3, 4, 5 in Section 3.2 is inappropriate, please correct it.
13. In line 237, “rm1” >>“rm”.
14. In Fig. 1, the disaster chain is not well expressed. I suggest the Figure should be improved.
15. In lines 69-70, the “foreign scholars [5-7] and domestic 69 scholars” is suggest to be deleted by only citing the reference.
16. The background of the 14 experts should be introduced.
Author Response
Dear reviewer,
Thanks for your letter and comments concerning our manuscript entitled” Resilience Evaluation of High-speed Railway Subgrade Construction System in Goaf Sites “(Manuscript ID: sustainability-1765837). Those comments are all valuable and helpful for revising and improving our paper. We have studied all comments carefully and have made conscientious corrections. I believe that the addressing of these comments has greatly improved the quality of this manuscript.
Responses to reviewer’ comments:
Question 1:Most of the manuscript is discussing the concept of resilience and a subjective evaluation system. The evaluation system is also personnel-dependent although both subjective weight and objective index are considered. It is not clear what is the core scientific problem of the manuscript. Is it the concept? Resilience index system? Or the evaluation system?
Answer :The core problem and ultimate goal of this paper is to evaluate the resilience of the construction system, but this purpose is inseparable from the introduction of the concept of resilience and the establishment of the evaluation system. First of all, previous studies on the resilience of high speed railway construction system above goaf area are insufficient, the concept of the resilience of the system should be described, and the specific connotation of this concept and its embodiment in the system should be clarified,so as to establish the resilience evaluation index system, and ultimately achieve the research results of this paper — the determination of the system resilience grade. This also provides the basis for the next resilience system optimization.
Question 2:The abstract is too long and there is no description on the establishment method of first grade indexes and second grade indexes.
Answer:We thank the reviewer for the valuable comments and suggestions. Regarding the establishment methods of first grade indexes and second grade indexes in the text, we have not described enough details before, so we have revised the abstract part according to your comments. (Abstract, page1)
Question 3: The introduction should be improved. Some references related to this article should be used.
Answer: Thanks for your constructive suggestions. In the introduction, we added some references related to the establishment of resilience index system and resilience evaluation method. (Introduction, page 4)
Question 4:What is new of this research? What is the major aim of the article?
Answer: We thank the reviewer for the valuable comments. The new discovery of this paper is that through consulting relevant literature on resilience evaluation, it is found that most of the previous studies on resilience evaluation focus on urban resilience, and most of the studies on the resilience of high-speed railway are also on the operation of high-speed railway, few studies introduce resilience into the high-speed railway subgrade construction system, and the role of goaf can be taken into account in the process of resilience evaluation. So our research object of resilience evaluation has improved compared with previous research objects, and ultimately we achieve the purpose of studying the resilience evaluation of high-speed railway construction system in goaf sites through the selection of resilience indexes and the establishment of resilience evaluation model.
Question 5. In section 2.1, According to the previous research results, the construction of high-speed railway subgrade in goaf sites is divided into organization member system, material technology system, management system, and environmental system. Corresponding references should be clearly indicated.
Answer: Thank you very much for your careful review work and helpful suggestions. According to your reminder, we have marked the references in the first paragraph of Section 2.1. (Page 5)
Question 6:The references in Table 2 are incorrectly formatted.
Answer: Thanks for the reviewer pointing out our mistake and providing us with helpful suggestions. This manuscript has been revised in accordance with the correct reference format, and it is worth noting that the original Table 2 has been deleted because it is too long, and the reference references cited are marked out in Table 1.(Page 7-8)
Question 7:English writing is generally good; some tenses need further confirmation.
Answer: Thank you very much for your careful review work and helpful suggestions. We are sorry for the carelessness in writing tenses, in the article, we have tried our best to find mistakes and modify, but there may be some shortcomings, we hope you can kindly comment.
Question8:The origin of resilience indexes should be more clearly explained.
Answer: We really appreciated the helpful suggestions. Indeed, in previous manuscripts, our description of the origin of the resilience indexes was not clear enough, so we added in section 2.2.3. (Page7-8)
Question 9:All formulas should be numbered.
Answer:Thanks for your kindly advice. We have numbered all the formulas in this article in accordance with your requirements and modified the number of the formulas referenced in this article
Question 10: There is an error in line 214.
Answer: In the new version, the paragraph at line 214 has been deleted, but we hope you can continue to point out our problems.
Question 11:The section number 1, 2, 3, 4, 5 in Section 3.2 is inappropriate, please correct it.
Answer :We are sorry for this mistake. According to your suggestion, we have changed the section number of section 3.2. (Page10-11)
Question 12. In line 237, “rm1” >>“rm”.
Answer: Thanks for your reminder, we have revised it.(Page11)
Question 13. In Fig. 1, the disaster chain is not well expressed. I suggest the Figure should be improved.
Answer: Thank for pointing out this important issue. I have modified the first paragraph of the introduction to the article on the unclear expression of the disaster chain and supplemented the Figure1. (Line3-7, page2; figue 1,page3)
Question14:In lines 69-70, the “foreign scholars [5-7] and domestic 69 scholars” is suggest to be deleted by only citing the reference.
Answer: Thank you for your suggestion. This sentence was marked in the paragraph 2 of the introduction. (Page3)
Question15:The background of the 14 experts should be introduced.
Answer: In order to take this concern into account, we have supplemented the background of 14 experts in the section (4) ‘Hydrogeological and climatic conditions’ of “Case applications” with your suggestion according to your comments.( Page12)
We are looking forward to your reply.
Round 2
Reviewer 2 Report
This manuscript can be accepted in current form.
Reviewer 3 Report
The authors have responded to my comments. I suggest the manuscript can be accepted.